# Compositional Neural Network Verification via Assume-Guarantee Reasoning

**Hai Duong**
George Mason University
Fairfax, VA 22030
hduong22@gmu.edu

**David Shriver**
Carnegie Mellon University
Pittsburgh, PA 15213
dshriver@andrew.cmu.edu

**ThanhVu Nguyen**
George Mason University
Fairfax, VA 22030
tvn@gmu.edu

**Matthew B. Dwyer**
University of Virginia
Charlottesville, VA 22904
matthewbdwyer@virginia.edu

## Abstract

Verifying the behavior of neural networks is necessary if developers are to confidently deploy them as parts of mission-critical systems. Toward this end, researchers have been actively developing a range of increasingly sophisticated and scalable neural network verifiers. However, scaling verification to large networks is challenging, at least in part due to the significant memory requirements of verification algorithms. In this paper, we propose an assume-guarantee compositional framework, CoVeNN, that is parameterized by an underlying verifier to generate a sequence of verification sub-problems to address this challenge. We present an iterative refinement-based strategy for computing assumptions that allow sub-problems to retain sufficient accuracy. An evaluation using 7 neural networks and a total of 140 property specifications demonstrates that CoVeNN can verify nearly 7 times more problems than state-of-the-art verifiers. CoVeNN is part of the NeuralSAT verification project: https://github.com/dynaroars/neuralsat.

## 1 Introduction

Machine learning (ML) techniques are advancing rapidly and have reached a level of performance across a range of challenging tasks, e.g., in the medical [1, 2] and autonomous driving [3, 4, 5] domains, that has led developers of critical systems to include ML models as components. To assure that such systems are fit for deployment, researchers have developed a variety of formal verification techniques to prove correctness properties of ML models, e.g., [6, 7, 8, 9, 10, 11, 12].

Advances in neural network verification (NNV) have been dramatic since the landmark paper by Katz et al. [6] which verified properties of models comprised of 6 linear layers. VNN-COMP [13, 14, 15, 16] has chronicled those advances by documenting the growth of benchmarks and verifiers. The largest benchmark in the competition, as measured by the number of layers, has grown from 6 to 21; where all but 2–3 of those layers are convolutional. Although these networks present are challenging for verifiers, they do not reflect the complexity of modern ML models.

While the number of layers in a network is not the only factor that contributes to the difficulty of a verification problem, it is directly related to its exponential complexity [6]. For SoTA verifiers, like $\alpha\beta$-CROWN [8], NeuralSAT [17], and PyRAT [18], the worst-case involves each layer generating multiple states which each serve as the starting point for verification of the suffix of the network from

that state forward. These verifiers perform a variety of optimizations to mitigate such state splitting, e.g., by tightening state encodings [19, 10], but complexity grows with the depth of the model.

This complexity is manifest both in increased runtime and, perhaps more importantly, in memory utilization. SoTA NNV tools make use of GPUs to efficiently manipulate high-dimensional abstractions of model states and GPU memory is generally more limited than CPU memory—in our evaluation (§4) GPU VRAM is limited to 24 GB. If verification requires more GPU memory than is available, then the verifier will abort with an out-of-memory (OOM) error. Fig. 1 shows the memory-consumption of the two top performing verifiers in VNN-COMP'24: $\alpha\beta$-CROWN and NeuralSAT, as they check 10 randomly generated local robustness properties of ResNet models trained on CIFAR10, with an increasing number of residual blocks in each model. The x-axis corresponds to the number of blocks within the model (e.g., 16 corresponds to ResNet-50 [20]). The y-axis plots the maximum memory consumed, as a percentage of 24 GB, by the verifier across the 10 verification problems. A point in the plot is shown if the verifier returns normally on any problem, regardless of whether the result is verified or unknown. $\alpha\beta$-CROWN and NeuralSAT are able to verify all problems up to 9 blocks, but beyond that they exhaust memory. While GPU memory has grown slowly over time, the pace of that growth cannot be relied on for scaling to large neural network verification problems.

In this paper, we introduce a framework for *Compositional Verification of Neural Networks* (CoVeNN) that can be parameterized by an underlying NNV tool, like $\alpha\beta$-CROWN and NeuralSAT. CoVeNN works by *decomposing a neural network into subnetworks* that are then verified independently. As depicted in Fig. 1, this allows CoVeNN to scale to larger networks than the underlying verifier.

The key to CoVeNN is the ability to encode the verification of the subnetworks as a series of *assume-guarantee* reasoning steps. After verifying each subnetwork, CoVeNN merges relevant verifier states, such as approximation bounds, into a compact summary that serves as the assumption for the next subnetworks. The corresponding *guarantee* ensures that, under this assumption, the subnetwork behaves correctly. Together, these assume-guarantee pairs are composed to establish the correctness of the full network.

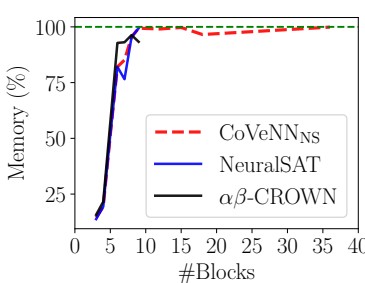

Fig. 1: Tools' scalability on ResNet-based instances. Maximum memory usage across 10 CIFAR10-based properties and ResNet-based networks with increasing numbers of blocks, comprised of 3 CNNs with ReLUs.

State merging is a classic approach for managing the cost of analysis and verification [21], but it risks introducing overapproximation that may prevent properties from being proven. To mitigate this, CoVeNN incorporates multiple refinement strategies that sharpen the precision of assumptions. As detailed in Tab. 4, CoVeNN matches the ability of underlying verifiers on problems for which they complete and the refinement strategies allow it to prove properties when scaling to much larger networks. CoVeNN does introduce overhead relative to the underlying verifier, but our evaluation (§4) on a set of challenging verification problems shows that substantial reduction in memory consumption translates to a significant increase in the ability to verify problems without an exorbitant time penalty.

**Related Work** NNV is still a relatively young field, and few lines of work have explored compositional NN verification [22, 23]. However, no prior technique can handle the scale or complexity of networks like RESNET36. We discuss these work and others in more detail in Apdx. D.

**Contributions** The primary contribution of this paper lies in the definition of a verifier-independent framework for compositional assume-guarantee verification of neural network properties. We implement and evaluate CoVeNN's ability to reduce verifier memory consumption and increase the number of properties proven (§4) and assess overall performance of CoVeNN relative to SoTA DNN verifiers on verification problems formulated over variants of neural network architectures.

## 2    Background

**DNN Verification** Verification of networks using piecewise-linear activation (e.g., ReLU) can be represented as a satisfiability problem [6, 11, 24, 10]. For an $L$-layer ReLU-based network $\mathcal{N}$ with

$N_l$ neurons in layer $l$, the formula:

$$\alpha \equiv \bigwedge_{i\in[1,L];\; j\in[1,N_l]} v_{i,j} = \max\Big( \sum_{k\in[1,N_l]} (w_{i-1,j,k} \cdot v_{i-1,k}) + b_{i,j}, 0 \Big)$$

defines the network. Given $\alpha$ and a property $\phi \equiv \phi_{in} \Rightarrow \phi_{out}$ a *DNN verification problem* is formulated by checking the satisfiability: $\alpha \wedge \phi_{in} \wedge \neg\phi_{out}$. If it is unsatisfiable, then $\phi$ is a valid property of $\mathcal{N}$. Otherwise, $\phi$ is not valid and a *counterexample*—a witness that $\phi$ is not valid—-is a satisfying assignment to the input variables in $\phi_{in}$.

**DNN Verifiers**  Modern DNN verifiers [18, 12, 11, 9, 10, 8, 24, 17, 19] adopt techniques from abstract interpretation [25] for efficiency. Since most properties studied in previous work can be expressed as a Boolean expression over a linear equation of $\mathcal{N}$'s outputs, where $\phi_{out}$ can be merged as the last layer of $\mathcal{N}$ to produce an objective function $f := \phi_{out} \circ \mathcal{N}$ [19, 9], the final goal reduces to prove: $\forall x \in \phi_{in} : f(x) \geq 0$. Solving $\min_{x\in\phi_{in}} f(x)$ is challenging due to the non-linearity of DNNs. Modern DNN verifiers overapproximate nonlinear computations of $\mathcal{N}$ to efficiently estimate the lower bound of $f(x)$, denoted as $lb$, i.e., $\forall x \in \phi_{in} : lb \leq f(x)$, then providing $lb \geq 0$ is sufficient to formally prove $f(x) \geq 0$. This allows abstraction-based DNN verifiers to side-step the disjunctive splitting that is the performance bottleneck of constraint-based DNN verifiers.

**Compositional Verification**  For more than four decades researchers have been investigating *compositional* methods to scale verification of complex systems [26]. One widely used compositional approach, termed *rely-guarantee* or *assume-guarantee* reasoning, was introduced by Stark [27]. For a system $M$ with a specification $\phi$, the goal is to prove $\phi_{in}, M \models \phi_{out}$—we denote such a proof goal with the triple $\langle \phi_{in}, M, \phi_{out} \rangle$. Compositional reasoning divides a system into parts, $\{M_1, \ldots, M_k\}$, and formulates a set of local verification problems $\langle A_i, M_i, G_i \rangle$ such that the guarantees of one component implies the assumptions of another and $\phi_{in}$ and $\phi_{out}$ are the assumption of the first and guarantee of the final components, respectively.

The promise of compositional methods is that they can reduce the complexity of verification by replacing reasoning about the product of the $M_i$ with reasoning about their sum. However, realizing such a framework is non-trivial: it requires suitable rules to relate the guarantees of one component to the assumptions of another, careful selection of decomposition strategies to achieve cost-effective verification [28], and the identification of appropriate assumptions $A_i$ [29]. CoVeNN addresses these challenges by exploiting the inherent sequential structure of neural architectures to define compositional proof rules tailored to layer-wise reasoning, adaptively selecting the degree of decomposition to maximize proof completion, and iteratively refining assumptions to support verification. Our approach builds on recent work in neural network verification by leveraging modern verifiers' ability to compute tight overapproximations of intermediate outputs, which we then use as assumptions for verifying subsequent layers.

## 3   Compositional Verification of Neural Networks

Alg. 1 shows the CoVeNN algorithm, which takes as input the DNN $\mathcal{N}$, the formulae $\phi_{in} \Rightarrow \phi_{out}$ representing the property to be proven, the factor $P$ indicating the number of generating assumptions, the scale factor $F$ for constructing assumptions, and the number of iterative rounds $r$. CoVeNN returns unsat if $\phi$ is a valid property of $\mathcal{N}$, and unknown otherwise.

CoVeNN consists of three main phases: (i) decomposing the original network into subnetworks (line 1), (ii) computing a coarse overapproximation of intermediate assumptions (line 2–line 4), then checking the last subnetwork (line 5) , and (iii) refining the assumptions iteratively until the problem can be verified (line 12–line 18). The following sections describe each phase in detail.

### 3.1   Decomposition

CoVeNN starts by splitting the original network into $K$ subnetworks $(\mathcal{N}_1, ..., \mathcal{N}_K)$ (line 1), where $K$ is internally inferred by CoVeNN using the heuristics described below. This step reduces verification complexity by enabling the sequential verification of smaller, more manageable subnetworks.

More formally, decomposition works as follows. A network, $\mathcal{N}$, is defined by a computation graph, $\mathcal{G}_\mathcal{N}$, whose nodes define computations, e.g., matrix multiplication, and whose edges describe data flow between computations. We restrict our attention to acyclic computation graphs, which are common

**Alg. 1:** `CoVeNN` algorithm.

---

**input** : Verifier $\mathcal{V}$, DNN $\mathcal{N}$, property $\phi_{in} \Rightarrow \phi_{out}$, number of neurons $P$, scale factor $F$ and rounds $r$
**output** : *unsat* if the property is valid and *unknown* otherwise

1   $(\mathcal{N}_1, ..., \mathcal{N}_K) \leftarrow$ `decomposeNetwork`$(\mathcal{N})$ // automatically split $\mathcal{N}$ into $K$ subnetworks
2   $\gamma_{over} \leftarrow [\phi_{in}]$
3   **for** $i \in [1, ..., K]$ **do** // initialize coarse overapproximation for each subnetwork
4     $\lfloor \quad \gamma_{over}.append($`overApproximate`$(\mathcal{N}_i, \gamma_{over}[i-1]))$
5   $lb \leftarrow \mathcal{V}.$`check`$(\gamma_{over}[K], \phi_{out})$ // check last subnetwork overapproximation w.r.t. output property
6   **if** $all(lb \geq 0)$ **then** // check if last subnetwork is verified
7     $\lfloor$ **return** *unsat* // $\langle \phi_{in}, \mathcal{N}, \phi_{out} \rangle$ is valid

8   **while** $r > 0$ **do**
9     **for** $i \in [1, ..., K-1]$ **do** // attempt to verify first $K-1$ subnetworks
10       $\gamma_{assume} \leftarrow$ `generateAssumption`$(\mathcal{N}_i, \gamma_{over}[i-1], \gamma_{over}[i], P, F)$ // see (Apdx. C)
11       $lb \leftarrow \mathcal{V}.$`verify`$(\mathcal{N}_i, \gamma_{over}[i-1], \gamma_{assume})$ // verify assumptions $\langle \gamma_{in}, \mathcal{N}_i, \gamma_{assume} \rangle$
12       $(L_{over}, U_{over}) \leftarrow \gamma_{over}[i]$ // extract bounds to refine (tighten)
13       **for** $\{(i, rhs, direction), li\} \in zip(\gamma_{assume}, lb)$ **do** // refine assumption for each neuron $i$-th
14         **if** $direction = $ " $\geq$ " **then** // assume $Y_i \geq rhs$, guarantee $Y_i - rhs \geq li$
15           $\lfloor \quad L_{over}[i] \leftarrow rhs + li$
16         **else** // assume $Y_i \leq rhs$, guarantee $-Y_i + rhs \geq li$
17           $\lfloor \quad U_{over}[i] \leftarrow rhs - li$
18       $\lfloor \quad \gamma_{over}[i] \leftarrow (L_{over}, U_{over})$ // update refined (tightened) bounds
19     $lb \leftarrow \mathcal{V}.$`verify`$(\mathcal{N}_K, \gamma_{over}[K-1], \phi_{out})$ // verify last subnetwork
20     **if** $all(lb \geq 0)$ **then** // check if last subnetwork is verified
21       $\lfloor$ **return** *unsat* // $\langle \phi_{in}, \mathcal{N}, \phi_{out} \rangle$ is valid
22     $r \leftarrow r - 1$
23 **return** *unknown*

---

in many classes of ML models. Our approach supports any decomposition into $k$ subnetworks such that: $\mathcal{N} := \mathcal{N}_k \circ \mathcal{N}_{k-1} \circ \ldots \circ \mathcal{N}_1$, and the input nodes of each $\mathcal{N}_i$ define a cut of $\mathcal{G}_\mathcal{N}$ [30]. Given such a decomposition a simple proof rule for sequential composition can be defined:

$$\frac{\langle A, \mathcal{N}_i, I \rangle \quad \langle I, \mathcal{N}_{i+1}, G \rangle}{\langle A, \mathcal{N}_{i+1} \circ \mathcal{N}_i, G \rangle} \tag{1}$$

For a $k$-way decomposed network, the rule is applied $k$ times with carefully chosen intermediate assumptions $I$, each serving as the guarantee of one step and the assumption of the next. If the first assumption is $A = \phi_{in}$ and the final guarantee is $G = \phi_{out}$, then verifying all subnetworks implies that the original network satisfies $\phi_{in} \Rightarrow \phi_{out}$.

**Decomposing Heuristics** A neural network can be decomposed into $k$ subnetworks in various ways, and the success of compositional reasoning depends largely on how it is decomposed [28]. `CoVeNN` uses four heuristics to automatically guide this choice. We prioritize (1) cuts that define the inputs of layers in the network, because this leads to subnetworks that have input/output shapes that are well-supported by existing verifiers; (2) minimum cuts, because these reduce the dimensionality of the intermediate assumption, $I$; (3) cuts that are *later* in the network – a cut, $c_2$, is later than cut, $c_1$, if all vertices in $c_2$ are dominated by some node in $c_1$, because this reduces imprecision in computation of $I$; and (4) cuts that yield the largest subnetworks that are amenable to verification by existing verifiers, because this minimizes the number of subnetworks that need to be verified.

### 3.2   Bound Approximations and Verification

`CoVeNN` next performs an initial, coarse *overapproximation* for each subnetwork sequentially (line 2–line 4). This is done using an off-the-shelf verifier $\mathcal{V}$—typically a modern BaB-based tool such as $\alpha\beta$-`CROWN`[8] or `NeuralSAT`[24]—which returns conservative output bounds based on the current property. `CoVeNN` leverages the computed bounds to verify properties and guide the iterative refinement discussed in §3.3.

Once the overapproximations are computed, `CoVeNN` simply checks whether the approximation of the last subnetwork satisfies the output property (line 5). As described in §2, if $\mathcal{V}$ returns a lower bound $lb \geq 0$ (line 6), the property is verified and `CoVeNN` concludes `unsat`. Otherwise, it proceeds with refinement to sharpen the result.

Although only the last subnetwork is being verified (since it directly relates to the original output property), this verification depends on the approximations computed for the earlier $K-1$ subnetworks. These serve as assumptions in a chain of *assume-guarantee* obligations. Each intermediate bound acts as both a guarantee for its originating subnetwork and an assumption for the next. We will discuss more about these assumptions and their refinement in the next section.

### 3.3 Iterative Refinement

In most cases, the initial approximation in §3.2 is too coarse to verify the property of interest. Thus, `CoVeNN` enters the *refinement* phase (line 9-line 18), which iteratively tightens the approximations until the problem can be verified (e.g., returns `unsat`) or `CoVeNN` exceeds the maximum predefined rounds $r$ (e.g., returns `unknown`). `CoVeNN`'s refinement has three main steps: (1) generating assumptions, (2) verifying assumptions, (3) refining assumptions. The algorithm makes up to $r$ iterations comprised of sequentially verifying each subnetwork.

**Generate Assumptions**   For each subnetwork $\mathcal{N}_i$, *assumptions* $\gamma_{assume}$ are generated (line 10) based on its input conditions and pre-computed overapproximation following the procedure in Alg. 2. `CoVeNN` automates the generation of assumptions by systematically interpolating between coarse overapproximations and tight sample-driven bounds on subnetwork outputs, as detailed in Apdx. C. These assumptions reflect the possible output behavior of $\mathcal{N}_i$ given the input property $\gamma_{in}$.

**Verify Assumptions**   `CoVeNN` attempts to verify $\gamma_{assume}$ of the subnetwork $\mathcal{N}_i$, or $\langle \gamma_{in}, \mathcal{N}_i, \gamma_{assume} \rangle$, using the verifier $\mathcal{V}$ (line 11). As described above, `CoVeNN` extracts the lower bound $lb$ from the verifier to facilitate this task. If verify an assumption fails (i.e., $lb < 0$), $\gamma_{assume}$ is invalid and `CoVeNN` refines them using verified $lb$ (line 12-line 18).

**Refine Assumptions**   This refinement adjusts the assumptions to eliminate unverified regions, making them hold for the current preconditions and subnetwork. Refined assumptions then are propagated forward serving as input property for the next subnetwork.

When an estimated assumption $\gamma_{assume}$ (line 10) cannot be verified, we need to refine it so that it becomes valid. Particularly, line 12-line 18 outlines `CoVeNN`'s refinement method, which adjusts $\gamma_{assume}$ using the formally verified lower bounds $lb$. `CoVeNN` first identifies the direction of the inequality to decide the appropriate refinement strategy. If the direction is "$\geq$", the assumption being verified is of the form $Y_i \geq rhs$ (line 14). The verifier $\mathcal{V}$ has only formally confirmed that $Y_i - rhs \geq li$, where $li < 0$, meaning that $Y_i$ is greater than or equal to $rhs$ adjusted by the lower bound $li$. Therefore, to make that assumption valid, the right-hand side value is loosened as $rhs + li$ (line 15). A dual of this process is used to refine the upper bounds.

### 3.4 Example

We illustrate `CoVeNN` by verifying that $\mathcal{N}$, depicted in Fig. 2a, has the property:

$$\phi \equiv \phi_{in} \implies \phi_{out} \equiv (-2 \leq x_1 \leq 2 \wedge -1 \leq x_2 \leq 1) \implies (y_1 > y_2) \tag{2}$$

When given the network $\mathcal{N}$ and the property $\phi$, `CoVeNN` first attempts to prove $\mathcal{N} \models \phi$, denoted by the triple $\langle \phi_{in}, \mathcal{N}, \phi_{out} \rangle$, using an underlying verifier $\mathcal{V}$. Suppose $\mathcal{V}$ fails to verify the property due to memory exhaustion.

`CoVeNN` now decomposes $\mathcal{N}$ into two subnetworks (line 1), $\mathcal{N}_1$ and $\mathcal{N}_2$, such that $\mathcal{N} = \mathcal{N}_2 \circ \mathcal{N}_1$ as shown in Fig. 2a. Next, `CoVeNN` uses $\mathcal{V}$ to compute an output overapproximation for the $K - 1$ subnetworks from the input condition $\phi_{in}$ (line 3–line 4). We call the computed constraint an *assumption* and use $\gamma_i$ to denote the assumption computed for network $i$. For this example, $\gamma_1 = -5 \leq n_{21} \leq 5 \wedge -10 \leq n_{22} \leq 10$. Since this assumption initially is an overapproximation, a consequence of this is that $\mathcal{V}$ has produced proof of $\langle \phi_{in}, \mathcal{N}_1, \gamma_1 \rangle$ inherently.

Once `CoVeNN` reaches the final subnetwork, it uses $\mathcal{V}$ to check the last overapproximation w.r.t. the output property (line 5), e.g., $\phi_{out}$. If this succeeds then we have a proof that $\langle \phi_{in}, \mathcal{N}, \phi_{out} \rangle$. In this case, this does not succeed, so `CoVeNN` attempts to refine the assumption (line 8-line 22).

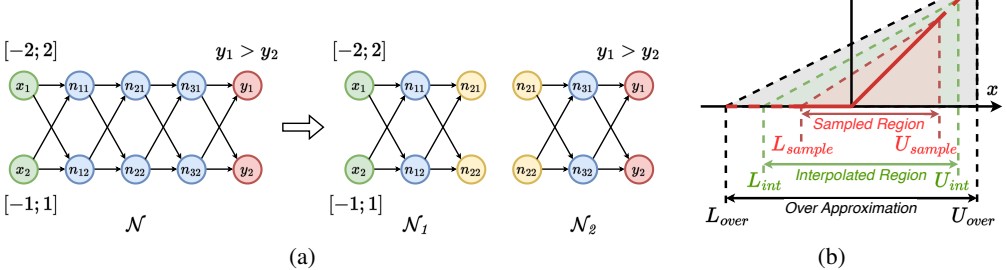

Fig. 2: (a) Example of a decomposed FC network with three hidden layers and (b) Regions of a hidden ReLU.

Refinement proceeds by sampling the behavior of $\mathcal{N}_1$, subject to $\phi_{in}$, and computes a hyperrectangle that tightly approximates sampled outputs; Fig. 2b depicts this sampled region in red. In the example, let this sampled assumption be $\sigma_1 = -2 \leq n_{21} \leq 2 \wedge -4 \leq n_{22} \leq 4$. Since this is a tight approximation of the sampled behavior the likelihood that it is a valid postcondition of $\mathcal{N}_1$ is low. We address this by interpolating between it and the overapproximating region to determine a new assumption (line 10), $\gamma_1'$ such that $\gamma_1 \supseteq \gamma_1' \supseteq \sigma_1$. With hyperrectangular constraints one approach is to simply scale the sampled region, $\gamma_1' = s \cdot (\gamma_1 - \sigma_1) + \sigma_1$, by some predefined factor, $s \in [0, 1]$. Fig. 2b depicts an interpolated region in green with a scaling factor of 0.5. In our example, this new assumption is $\gamma_1' = -3 \leq n_{21} \leq 3 \wedge -5 \leq n_{22} \leq 5$ and we use $\mathcal{V}$ to verify $\langle \phi_{in}, \mathcal{N}_1, \gamma_1' \rangle$ (line 11).

If this succeeds, then CoVeNN uses $\mathcal{V}$ to attempt to verify $\langle \gamma_1', \mathcal{N}_2, \phi_{out} \rangle$ (line 19). If it fails we exploit the output bounds computed by $\mathcal{V}$ to generate a valid assumption (line 12–line 18): $\gamma_1'' = (-4 \leq n_{21} \leq 4 \wedge -7 \leq n_{22} \leq 7)$ and CoVeNN then seeks to verify $\langle \gamma_1'', \mathcal{N}_2, \phi_{out} \rangle$ (line 19). In this example that verification succeeds, thereby completely the proof of $\langle \phi_{in}, \mathcal{N}, \phi_{out} \rangle$ through a sequence of simpler verification problems.

### 3.5 Formal Correctness

The correctness of CoVeNN is based on the soundness of the assume-guarantee decomposition (§3.1) and the iterative refinement of the assumptions (§3.3). Note that we assume that the underlying verifier $\mathcal{V}$ is sound, i.e., its overapproximations are valid (§3.2). Below we provide the theorems and proof sketches for the soundness of CoVeNN, the full proofs are provided in Apdx. B.

**Compositional Verification**   The following states that the chain of assumptions and guarantees of subnetworks (§3.1) proves the original property $\phi_{in} \Rightarrow \phi_{out}$ of the entire network $\mathcal{N}$.

**Thm. 1** (Compositional Verification via Assume-Guarantee Reasoning). *Given a neural network $\mathcal{N} : \mathbb{R}^n \to \mathbb{R}^m$ decomposed into $K$ subnetworks such that $\mathcal{N} = \mathcal{N}_K \circ \cdots \circ \mathcal{N}_1$, a property $\phi \equiv \phi_{in} \Rightarrow \phi_{out}$, and intermediate predicates $\gamma_1, \ldots, \gamma_{K-1}$, where $\gamma_0 = \phi_{in}$ and $\gamma_K = \phi_{out}$, the global specification holds whenever the every local property is valid:*

$$\left( \bigwedge_{k=1}^{K} \langle \gamma_{k-1}, \mathcal{N}_k, \gamma_k \rangle \right) \implies \langle \phi_{in}, \mathcal{N}, \phi_{out} \rangle$$

*Proof Sketch.* For each pair of adjacent networks as shown in Eq. 1, assume-guarantee obligations are proved, establishing that each intermediate predicate $\gamma_k$ is preserved under corresponding subnetwork $\mathcal{N}_k$. Composing these local guarantees forms a chain from the input property $\phi_{in}$ to the output property $\phi_{out}$, showing that the global specification holds for the entire network. $\square$

**Refinement Process**   The following states that the refinement process (§3.3) states that each refinement step always results in bounds that are formally valid, thereby ensuring that the iterative tightening of assumptions preserves correctness throughout the verification chain.

**Thm. 2** (Soundness of Iterative Bound Refinement). *Suppose the base verifier $\mathcal{V}$ soundly establishes, for a neuron $Y_i$, a right-hand side $rhs$, a direction $\preccurlyeq \in \{\geq, \leq\}$, and a corresponding verifier bound*

Tab. 1: Benchmark instances.

| Benchmark | Layers | Neurons | Parameters | Instances (U/S/?) |
|-----------|--------|---------|------------|-------------------|
| VAE_BASE | 20 | 43K | 10K | 20 / 0 / 0 |
| VAE_WIDE | 20 | 86K | 39K | 20 / 0 / 0 |
| VAE_DEEP | 28 | 44K | 15K | 7 / 0 / 13 |
| RESNET6 | 20 | 283K | 113K | 11 / 0 / 9 |
| RESNET12 | 38 | 627K | 230K | 9 / 0 / 11 |
| RESNET18 | 56 | 700K | 348K | 18 / 0 / 2 |
| RESNET36 | 110 | 1032K | 706K | 13 / 0 / 7 |
| **Total** | | | | **98 / 0 / 42** |

$\delta$ *(where $\delta < 0$ for "$\geq$" and $\delta > 0$ for "$\leq$"), that $Y_i - rhs \preccurlyeq \delta$. Then, the refined inequality*

$$Y_i \preccurlyeq (rhs + \delta)$$

*is verified. Therefore, updating the right-hand side to $rhs + \delta$ yields a verified assumption by $\mathcal{V}$.*

*Proof Sketch.* Given that the verifier soundly proves $Y_i - rhs \preccurlyeq \delta$, it follows that $Y_i \preccurlyeq (rhs + \delta)$ holds. Updating the bound accordingly yields a refined assumption that remains sound. □

**Soundness of `CoVeNN`**   The following combines the previous two theorems to show that `CoVeNN` is sound: if all local subproblems generated during decomposition are either formally verified or refined, then the global property holds for the original network.

**Thm. 3** (Soundness of `CoVeNN`). *Let $\mathcal{N}$ be a neural network and $\phi_{in}, \phi_{out}$ be input/output properties such that `CoVeNN` verifies $\mathcal{N}$ satisfies $\phi_{in} \implies \phi_{out}$. `CoVeNN` applies Thm. 1 to decompose $\mathcal{N}$ and assume all local subproblems are formally verified by a sound underlying verifier $\mathcal{V}$ or formally refined as Thm. 2. Then $\mathcal{N}$ indeed satisfies $\phi_{in} \implies \phi_{out}$.*

*Proof Sketch.* `CoVeNN` decomposes the network into subnetworks and verifies a sequence of assume-guarantee obligations using a sound verifier. By composing these verified local implications, the global property $\phi_{in} \Rightarrow \phi_{out}$ follows. □

**Non-Linear Activations**   While our discussion has primarily focused on ReLU activations, `CoVeNN` extends naturally to networks with other non-linear activations such as tanh and sigmoid. The support for non-ReLU is specific to the underlying verifier that we use to obtain bounds, e.g., $\alpha\beta$-`CROWN` and `NeuralSAT` both support non-ReLU activations via abstract interpretation techniques. `CoVeNN` extracts interval bounds from these verifiers as intermediate assumptions rather than discrete neuron status (e.g., on/off). These interval bounds remain valid regardless of the activation function and preserve soundness during refinement.

## 4   Evaluation

We evaluate the scalability and cost-effectiveness of `CoVeNN` based on three research questions on `CoVeNN`'s performance compared to state-of-the-art verifiers (**RQ1**); the effectiveness of refinement (§3.3) (**RQ2**); `CoVeNN`'s robustness to variations in the underlying verifier (**RQ3**); and the impact of parameter tuning on `CoVeNN`'s performance (**RQ4**).

**Underlying Verifiers**   We experiment with two variants of `CoVeNN`, each configured with a different underlying verifier: `NeuralSAT` and $\alpha\beta$-`CROWN`. Both tools are state-of-the-art in DNN verification[1], and allow us to extract the lower bound estimates needed for `CoVeNN`'s refinement process.

**Benchmarks**   We use two scalable families of benchmarks (Apdx. E). Tab. 1 provides details on the variants of the ResNet and VAE benchmarks used in our experiments. For each network, we generated 20 robustness properties. For ResNets these are local robustness classification properties and for VAEs these are local reconstruction robustness properties (Apdx. A). Across the 140 combinations

---

[1]See results in the VNN-COMP'24 report [15, Tab. 35]. `PyRAT` is commercial and has no available code.

Tab. 2: Comparing `CoVeNN` to SoTA verifiers; most solved problems in bold.

| Verifier | VAE_BASE | | | VAE_WIDE | | | VAE_DEEP | | | RESNET6 | | | RESNET12 | | | RESNET18 | | | RESNET36 | | | Overall | |
|---|---|---|---|---|---|---|---|---|---|---|---|---|---|---|---|---|---|---|---|---|---|---|---|
| | V | % | K | V | % | K | V | % | K | V | % | K | V | % | K | V | % | K | V | % | K | V | % |
| CoVeNN$_{NS}$ | **20** | 100.0 | 1-2 | **19** | 95.0 | 2 | **5** | 25.0 | 2 | **11** | 55.0 | 1 | **9** | 45.0 | 2 | **18** | 90.0 | 2 | **13** | 65.0 | 3 | **95** | 67.9 |
| CoVeNN$_{\alpha\beta}$ | 4 | 20.0 | 2 | - | - | - | **6** | 30.0 | 2 | 10 | 50.0 | 1 | **9** | 45.0 | 2 | 17 | 85.0 | 2 | **13** | 65.0 | 3 | 59 | 42.1 |
| CoVeNN$_{Refine}$ | 3 | 15.0 | 1 | - | - | - | - | - | - | **11** | 55.0 | 1 | **9** | 45.0 | 2 | 9 | 45.0 | 2 | **13** | 65.0 | 3 | 45 | 32.1 |
| NeuralSAT | 3 | 15.0 | 1 | - | - | - | - | - | - | **11** | 55.0 | 1 | - | - | - | - | - | - | - | - | - | 14 | 10.0 |
| $\alpha\beta$-CROWN | 1 | 5.0 | 1 | - | - | - | - | - | - | 10 | 50.0 | 1 | - | - | - | - | - | - | - | - | - | 11 | 7.9 |

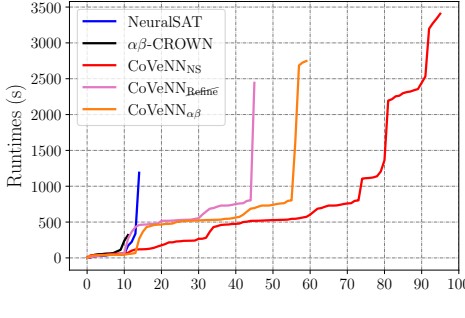

(a) Solved problems sorted by runtime.

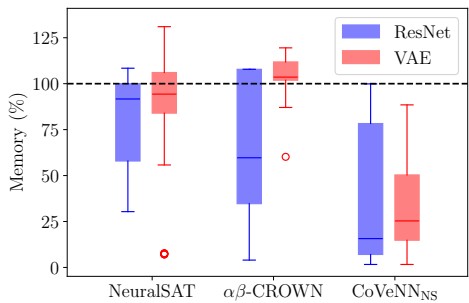

(b) Average memory usage per verifier.

Fig. 3: `CoVeNN` performance compared to SoTA verifiers.

of networks and properties: 98 are known to be `unsat` (U), none of them are `sat` (S), and of the remaining 42 instances no verifier in our study was able to solve the problem (?). Note that robustness properties can vary significantly in complexity based on the centerpoint and $\epsilon$, e.g., [31, Fig. 2].

**Setup** Our experiments were run on a Linux machine with an AMD Ryzen Threadripper PRO 5975WX 32-Core, 128 GB RAM, and an NVIDIA GeForce RTX 4090 GPU with 24 GB VRAM. Timeout for a single instance is set to 3600 seconds or the maximum number of rounds $r$ is 4. Note that our results are deterministic and no random process is involved. We describe detailed configuration and model information for our experiments in Apdx. E.

## 4.1 RQ1: Comparing to Non-Compositional SoTA Verifiers

Tab. 2 presents the results of running both `CoVeNN` variants (`CoVeNN`$_{NS}$ and `CoVeNN`$_{\alpha\beta}$), `CoVeNN`$_{Refine}$— a *naïve* version of `CoVeNN` without refinement using the `NeuralSAT` backend, and the standalone `NeuralSAT` and $\alpha\beta$-`CROWN` verifiers on the benchmarks. Column **V** shows the number of problems verified with the percentage solved shown in column **%**. Column **K** shows the number of decompositions inferred by `CoVeNN`. Tools that run out of memory or time out on a benchmark are indicated with a "-" (e.g., `NeuralSAT` cannot solve any instances of RESNET12 and VAE_DEEP, etc.). Across the benchmarks `CoVeNN` solves more than 6 times the number of problems than the best non-compositional solver. We note that neither `NeuralSAT` nor $\alpha\beta$-`CROWN` could solve any instances of VAE_WIDE, VAE_DEEP, RESNET12 and beyond, which demonstrates the ability of `CoVeNN` to scale verification beyond the state-of-the-art. RESNET36, which is comprised of 110 convolutional layers, requires the most aggressive decomposition ($K = 3$), but even for such a large network `CoVeNN` is able to verify 65% of the properties. `CoVeNN`$_{Refine}$ performs better than standalone verifiers, but falls short of any `CoVeNN` variants, demonstrating the importance of refinement (§4.2).

Fig. 3a and Fig. 3b provide additional details on runtime and memory usage. Fig. 3a shows that regardless of the underlying solver or whether `CoVeNN` uses its refinement strategy (§3.3) it can solve many more problems than `NeuralSAT` and $\alpha\beta$-`CROWN` within the same time constraints. While `NeuralSAT` and $\alpha\beta$-`CROWN` reach their limits after solving 14 problems, `CoVeNN` solves as many as 95 instances. Fig. 3b shows that `CoVeNN` consumes significantly less memory[2] than `NeuralSAT`

---

[2]Measured by the internal function `torch.cuda.mem_get_info` in PyTorch. Detailed calculation in Eq. 10.

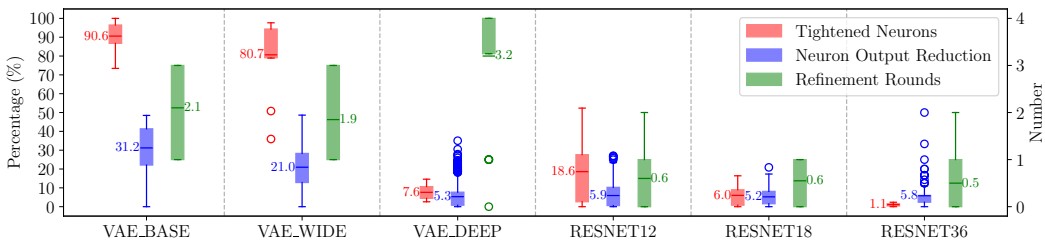

Fig. 4: Percentage of neurons tightened and percentage of their output range reduction.

and $\alpha\beta$-CROWN, which often encounter OOM errors. §4.4 reports on a more detailed parameter and ablation study of CoVeNN on these benchmarks.

## 4.2 RQ2: On the Effectiveness of Assumption Refinement

Fig. 3a shows that even without refinement, CoVeNN$_{\text{Refine}}$ solves nearly 4 times as many problems as SoTA methods and that this rises to nearly 7 times with refinement enabled. To explore how refinement achieves this we recorded additional data on the refinement process. Fig. 4 reports, for each problem where $K > 1$, the percentage of neurons that were tightened in some round of refinement (red); the percentage by which the output ranges of those neurons were reduced (blue); and number of rounds of refinement performed (green - on the 2nd y-axis).

For the VAE benchmarks, 17 of the BASE problems and all of the other problems involved refinement. There were more rounds of refinement and, except for the DEEP benchmark, this resulted in a high percentage of tightened neurons and significant output range reduction. For the DEEP benchmark, rounds of refinement continued until the timeout was reached on 15 of the 20 problems. For the RESNET benchmarks, we observe a very different profile. Fewer rounds of refinement were needed, especially for RESNET18 which was able to prove 18 of the 20 problems. This is more than double the amount that could be proven with refinement disabled. The verification problems are randomly sampled across all benchmarks which leads to different performance profiles. The data for RESNET12 and RESNET36 demonstrate that for some problems the compositional nature of CoVeNN alone can lead to verification improvements (see the CoVeNN$_{\text{Refine}}$ row in Tab. 2), but the additional effort of refinement does not yield further improvements.

## 4.3 RQ3: Robustness to Underlying Verifiers

Tab. 2 clearly indicates that CoVeNN significantly increase the number of solved problems regardless of the underlying verifier. While CoVeNN$_{\text{NS}}$ outperforms CoVeNN$_{\alpha\beta}$, our analysis suggests that this is not a fundamental limitation of either CoVeNN or $\alpha\beta$-CROWN.

We analyzed the performance of CoVeNN$_{\alpha\beta}$ on VAE_BASE and VAE_WIDE, the cases where there was a substantial performance gap. We found that $\alpha\beta$-CROWN does not support ConvTranspose layers in several of its heuristics for applying optimizations. This means that its performance in both verifying assumptions line 11 and verifying the final subnetwork line 19 suffer. We conjecture that better support for this layer type could ameliorate this issue, but we acknowledge that $\alpha\beta$-CROWN has numerous hyperparameters and we did not perform the type of expert tuning that the developers of $\alpha\beta$-CROWN apply when running benchmarks. The reduced performance on VAE benchmarks could be due to this as well.

The RESNET benchmarks show a clearer trend. $\alpha\beta$-CROWN comes with hyperparameter settings for this architecture which we reuse, and we see very consistent degrees of improved scalability regardless of the underlying verifier.

## 4.4 RQ4: CoVeNN's Performance with Parameter Tuning

Fig. 5 compares CoVeNN's performance across different parameter configurations using NeuralSAT as the underlying verifier. We explore variations in $P$ (number of assumptions per round) and $F$ (interpolation factor) against baselines including NeuralSAT, $\alpha\beta$-CROWN, and CoVeNN$_{\text{Refine}}$.

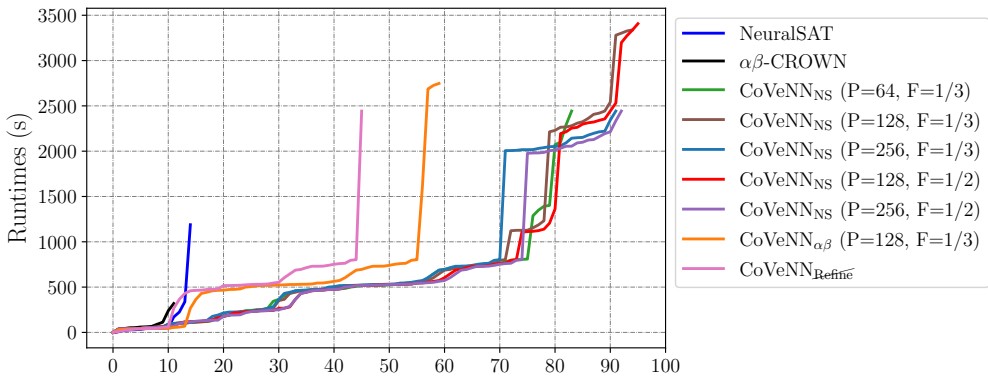

Fig. 5: Ablation study on different parameters of `CoVeNN`.

`CoVeNN` exhibits dependencies on $P$, which determines how many neurons are selected for refinement in each round. When $P$ is small ($P = 64$), `CoVeNN` refines fewer neurons per iteration, limiting its ability to tighten bounds and solving only 83 problems. Conversely, when $P$ is large ($P = 256$), `CoVeNN` attempts to refine many neurons simultaneously, increasing computational overhead and often causing timeouts before properties can be proven. The optimal $P = 128$ strikes a balance between refinement coverage and computational efficiency.

In contrast, `CoVeNN` demonstrates robustness to the interpolation factor $F$. The performance difference between $F = 1/3$ and $F = 1/2$ is marginal (e.g., a single problem when $P = 128$), indicating that `CoVeNN`'s iterative refinement process can effectively compensate for initially coarse assumptions regardless of the interpolation strategy.

> In summary, (**RQ1**) `CoVeNN` verifies over 6x more problems than SOTA verifiers and scales to deeper networks with less memory, highlighting the effectiveness of decomposition. (**RQ2**) Refinement *significantly boosts performance*: `CoVeNN`Refine already outperforms SoTA by 4x, and refinement raises this to nearly 7x. (**RQ3**) `CoVeNN` significantly improves verification regardless of underlying solvers (performance gaps depend on the backends, not from `CoVeNN` itself). (**RQ4**) `CoVeNN`'s performance depends on the number of neurons selected for refinement but is robust to the interpolation factor.

## 5  Conclusion

NNV has scaled tremendously in recent years, but networks continue to grow in complexity which limits the applicability of SoTA NNV techniques to real-world networks. Our work on `CoVeNN` is a first step in realizing the promise of compositional neural network verification, and preliminary results show that `CoVeNN` can scale significantly beyond SoTA verifiers.

**Limitations**  This work focuses on compositional verification, not *falsification*. Compositional falsification is more difficult because over-approximation favors verification rather than finding counterexamples. However, `CoVeNN` can potentially incorporate backward analysis methods [32] to compute under-approximations of inputs reachable from the violated output space. If the under-approximated input region is a subset of the specified precondition, the violation can be confirmed.

`CoVeNN` supports acyclic networks, e.g., feedforward structures, by finding appropriate "cuts" in the computation graph. For RNNs, `CoVeNN` can use unrolling [33] to create natural decomposition points after each timestep. For general cyclic graphs, handling cycles requires more sophisticated circular assume-guarantee proof rules [34].

Finally, `CoVeNN` does not use feedback from counterexamples, but we plan to use them to identify the dimensions within an assumption that must be tightened to eliminate violations.

**Potential Negative Societal Impact**  The techniques developed in this work can reveal flaws in sensitive applications that may be exploited for malicious intent. However, these same methods also allow developers to identify such attacks and fix them prior to model deployment.

## Acknowledgments and Disclosure of Funding

We thank the anonymous reviewers for their helpful comments. This work was supported in part by funds provided by the National Science Foundation awards 2129824, 2217071, 2501059, 2422036, 2319131, 2238133, and 2200621, and by an Amazon Research Award and an NVIDIA Academic Grant.

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

## A  DNN Properties

For a neural network, $\mathcal{N} : \mathbb{R}^n \mapsto \mathbb{R}^m$, specifications of *functional* properties constrain the output of $\mathcal{N}$ based on its input. Specifying such necessary conditions for DNN input-output relations is a topic of great interest with applications as test oracles in DNN testing [35], training objectives to maximize network property conformance [36], and biasing generative models to produce outputs that conform to a property [37]. A broad class of properties can be formulated using pairs of half-space polytopes – each specified as the conjunction of cutting planes – where one set defines the pre-condition, $\phi_{in}$, and another the post-condition, $\phi_{out}$. A variety of different properties can be formulated using such pre-post condition pairs but we focus on two popular classes of properties below.

**Classification Robustness**    A well-studied class of properties express local robustness properties of the form:

$$\forall p \in [0, \epsilon] : \mathcal{N}(x) = \mathcal{N}(x \pm p),$$

where $x$ is a specific seed input, often chosen from the held out test dataset. These express that network inference is invariant for perturbations within the convex $\epsilon$-ball, $x \pm p$, for the seed input. A variant of this, with a post-condition of the form $\|\mathcal{N}(x) - \mathcal{N}(x \pm p)\| \leq \beta$ can be defined for regression networks.

**Reconstruction Robustness**    Generative models are formulated as neural networks, $\mathcal{N} : \mathbb{R}^n \mapsto \mathbb{R}^n$, and are trained to approximate the identify function [38, 39]. For such, models one can adapt the regression robustness property above, to define a local reconstruction robustness property of the form:

$$\forall p \in [0, \epsilon] : \|\mathcal{N}(x \pm p) - x\|_\infty \leq \beta,$$

for a seed input $x$. The $\beta$ parameter expresses the expected agreement between an input and its reconstruction.

## B  Correctness Proofs of `CoVeNN`

Let $\mathcal{N} : \mathbb{R}^n \to \mathbb{R}^m$ denote a neural network, and let $\phi_{in}(x)$ and $\phi_{out}(y)$ be input and output property of interest, respectively. The global verification objective is to prove:

$$\forall x \in \mathbb{R}^n : \; \phi_{in}(x) \implies \phi_{out}(\mathcal{N}(x)) \tag{3}$$

**Chain of Proofs.**    Given $\mathcal{N}$, we decompose into $K$ sequential subnetworks: $\mathcal{N} = \mathcal{N}_K \circ \mathcal{N}_{K-1} \circ \cdots \circ \mathcal{N}_1$, where each $\mathcal{N}_k : \mathbb{R}^{d_{k-1}} \to \mathbb{R}^{d_k}$, with $d_0 = n$ and $d_K = m$.

Let $\gamma_0 := \phi_{in}$ and $\gamma_K := \phi_{out}$; we seek to synthesize $K - 1$ intermediate predicates (assumptions) $\gamma_1, \ldots, \gamma_{K-1}$ over intermediate network states.

Following the classical assume-guarantee (sequential composition) proof principle, the global property is entailed by the following chain of $K$ local obligations:

$$\forall k = 1, \ldots, K : \qquad \langle \gamma_{k-1}, \mathcal{N}_k, \gamma_k \rangle \tag{4}$$

If all these obligations hold, we have:

$$\left( \bigwedge_{k=1}^{K} \langle \gamma_{k-1}, \mathcal{N}_k, \gamma_k \rangle \right) \implies \langle \phi_{in}, \mathcal{N}, \phi_{out} \rangle \tag{5}$$

Thus global verification is reduced to a finite sequence of smaller verification problems.

**Automated Synthesis of Intermediate Assumptions.**    Given a subnetwork $\mathcal{N}_k$, input constraint $\gamma_{in}$, an overapproximate output bound $\gamma_{over} = (L_{over}, U_{over})$, number of neurons $P$, and scale factor $F \in [0, 1]$, we synthesize the candidate assumption $\gamma_{assume}$ as follows.

First, `CoVeNN` computes the sampled output region by solving:

$$L_{sample} = \min_{x \in \gamma_{in}} \mathcal{N}_k(x) \qquad U_{sample} = \max_{x \in \gamma_{in}} \mathcal{N}_k(x) \tag{6}$$

where each minimization and maximization is performed through backpropagation.

Next, `CoVeNN` interpolates between the overapproximate and sampled bounds to obtain:

$$L_{int} = L_{sample} - F \cdot (L_{sample} - L_{over}) \qquad U_{int} = F \cdot (U_{over} - U_{sample}) + U_{sample} \qquad (7)$$

Let $Select(L_{int}, U_{int}, P)$ denote a selection of $P$ neuron indices for which the interval $[L_{int}[i], U_{int}[i]]$ is top-$P$ largest. The synthesized assumption $\gamma_{assume}$ for $\mathcal{N}_k$ is then the set of inequalities:

$$\gamma_{assume} = \left\{ Y_i \geq L_{int}[i], \quad Y_i \leq U_{int}[i] \; \middle| \; i \in Select(L_{int}, U_{int}, P) \right\} \qquad (8)$$

where $Y_i$ denotes the $i$-th output of $\mathcal{N}_k$.

**Iterative Refinement.** For each refinement round $r$, and for each subnetwork $\mathcal{N}_i$ ($i = 1, \ldots, K-1$), let the current overapproximate bound for its output be denoted by $(L_i^{(r)}, U_i^{(r)})$. Given an assumption $\gamma_{\text{assume}}$ (constructed as above) and the formally verified lowerbounds $lb$ extracted from the underlying verifier, consider all neurons $j$ selected for tightening. For each such $j$ with assumption direction ("$\geq$" or "$\leq$") and associated right-hand side $rhs$, let $l_j$ be the corresponding lower bound from $lb$. The refined bounds are updated by:

$$L_i^{(r+1)}[j] = \begin{cases} rhs + l_j, & \text{if direction is "}\geq\text{"} \\ L_i^{(r)}[j], & \text{otherwise} \end{cases} \quad U_i^{(r+1)}[j] = \begin{cases} rhs - l_j, & \text{if direction is "}\leq\text{"} \\ U_i^{(r)}[j], & \text{otherwise} \end{cases} \quad (9)$$

All other coordinates are left unchanged. The updated tuple $(L_i^{(r+1)}, U_i^{(r+1)})$ forms the new tightened overapproximation for the next refinement round, and subsequent rounds proceed analogously unless the verification succeeds or the allowed number of rounds is exhausted.

## B.1 Chain of Proofs

**Thm. 1** (Compositional Verification via Assume-Guarantee Reasoning) *Given a neural network $\mathcal{N} : \mathbb{R}^n \to \mathbb{R}^m$ decomposed into $K$ subnetworks such that $\mathcal{N} = \mathcal{N}_K \circ \cdots \circ \mathcal{N}_1$, a property $\phi \equiv \phi_{in} \Rightarrow \phi_{out}$, and intermediate predicates $\gamma_1, \ldots, \gamma_{K-1}$, where $\gamma_0 = \phi_{in}$ and $\gamma_K = \phi_{out}$, the global specification holds whenever every local property is valid:*

$$\left( \bigwedge_{k=1}^{K} \langle \gamma_{k-1}, \mathcal{N}_k, \gamma_k \rangle \right) \implies \langle \phi_{in}, \mathcal{N}, \phi_{out} \rangle$$

*Proof.* Let $\mathcal{N} = \mathcal{N}_K \circ \mathcal{N}_{K-1} \circ \cdots \circ \mathcal{N}_1$ denote the decomposition into $K$ composed subnetworks. Let the intermediate predicates be $\gamma_0, \gamma_1, \ldots, \gamma_K$ with $\gamma_0 = \phi_{in}$ and $\gamma_K = \phi_{out}$. We are given that for every $k = 1, \ldots, K$,

$$\forall z \in \mathbb{R}^{d_{k-1}} : \gamma_{k-1}(z) \implies \gamma_k(\mathcal{N}_k(z)), \qquad \text{or} \qquad \langle \gamma_{k-1}, \mathcal{N}_k, \gamma_k \rangle$$

We show that:

$$\forall x \in \mathbb{R}^n : \phi_{in}(x) \implies \phi_{out}(\mathcal{N}(x)), \qquad \text{or} \qquad \langle \phi_{in}, \mathcal{N}, \phi_{out} \rangle$$

Let $x \in \mathbb{R}^n$ such that $\phi_{in}(x)$ holds. That is, $\gamma_0(x)$.

- **Base case** ($k = 1$): By the first obligation, $\gamma_0(x) \implies \gamma_1(\mathcal{N}_1(x))$, so $\gamma_1(z_1)$ holds for $z_1 = \mathcal{N}_1(x)$.

- **Induction step:** Assume for some $1 \leq k < K$ that $\gamma_{k-1}(z_{k-1})$ holds for some $z_{k-1}$. Then, by the $k$-th obligation, $\gamma_k(\mathcal{N}_k(z_{k-1}))$ holds for $z_k = \mathcal{N}_k(z_{k-1})$.

By unrolling, starting from $\gamma_0(x)$, we obtain via repeated application:

$$\gamma_1(z_1), \; \gamma_2(z_2), \; \ldots, \; \gamma_K(z_K), \text{ where } z_k = \mathcal{N}_k(z_{k-1}), \; z_0 = x$$

In this case, $z_K = \mathcal{N}_K(\cdots \mathcal{N}_1(x)) = \mathcal{N}(x)$ and $\gamma_K(z_K)$ means $\phi_{out}(\mathcal{N}(x))$.

Therefore, for any $x$ satisfying $\phi_{in}(x)$, we have $\phi_{out}(\mathcal{N}(x))$. That is,

$$\forall x : \phi_{in}(x) \implies \phi_{out}(\mathcal{N}(x))$$

Thus, the global specification holds. $\qquad\qquad \square$

**Alg. 2:** Generate assumptions.

**input** : DNN $\mathcal{N}$, input property $\gamma_{in}$, overapproximation $\gamma_{over}$, number of neurons $P$, and scale factor $F$
**output** : Assumptions $\gamma_{assume}$

1   $(L_{over}, U_{over}) \leftarrow \gamma_{over}$
2   $L_{sample} \leftarrow \texttt{Minimize}(\mathcal{N}, \gamma_{in})$ // minimize output of $\mathcal{N}$ for samples from $\gamma_{in}$
3   $U_{sample} \leftarrow \texttt{Maximize}(\mathcal{N}, \gamma_{in})$ // maximize output of $\mathcal{N}$ for samples from $\gamma_{in}$
4   $L_{int} \leftarrow L_{sample} - F \cdot (L_{sample} - L_{over})$ // interpolate output lower bounds of $\mathcal{N}$
5   $U_{int} \leftarrow F \cdot (U_{over} - U_{sample}) + U_{sample}$ // interpolate output upper bounds of $\mathcal{N}$
6   $\gamma_{assume} \leftarrow [\,]$
7   **for** $i \in \texttt{Select}(L_{int}, U_{int}, P)$ **do** // select top-$P$ neurons
8       $\gamma_{assume}.append((i, L_{int}[i], \geq))$ // tighten assumption $Y_i \geq L_{int}[i]$
9       $\gamma_{assume}.append((i, U_{int}[i], \leq))$ // tighten assumption $Y_i \leq U_{int}[i]$
10   **return** $\gamma_{assume}$

## B.2   Soundness of Iterative Bound Refinement

**Thm. 2** (Soundness of Iterative Bound Refinement) *Suppose the base verifier $\mathcal{V}$ soundly establishes, for a neuron $Y_i$, a right-hand side $rhs$, a direction $\preccurlyeq \in \{\geq, \leq\}$, and a corresponding verifier bound $\delta$ (where $\delta < 0$ for "$\geq$" and $\delta > 0$ for "$\leq$"), that $Y_i - rhs \preccurlyeq \delta$. Then, the refined inequality*

$$Y_i \preccurlyeq (rhs + \delta)$$

*is soundly verified. Therefore, updating the right-hand side to $rhs + \delta$ yields an assumption formally verified by $\mathcal{V}$.*

*Proof.* Consider the case when the direction is "$\geq$". The assumption being verified takes the form $Y_i \geq rhs$ (line 14). $\mathcal{V}$ soundly verifies that $Y_i - rhs \geq \delta$, where $\delta = li$ and $\delta < 0$. This implies $Y_i \geq rhs + \delta$. Therefore, to ensure this assumption is valid, we update the right-hand side to $rhs + \delta$ (line 15), confirming $Y_i \geq rhs + \delta$ holds. Similarly, a dual of this process is used to refine the upper bounds. In both cases, replacing the original bound $rhs$ with $rhs + \delta$ yields an assumption that is formally validated by the verifier. Thus, the refinement procedure produces sound bounds. $\qquad\square$

This theorem formalizes the soundness of our bound refinement procedure, which is a critical component of our iterative assumption generation. Specifically, it ensures that each refinement step, guided by the verifier's output, always results in bounds that are formally valid, thereby guaranteeing that the iterative tightening of assumptions preserves correctness throughout the verification chain.

## B.3   Soundness of `CoVeNN`

**Thm. 3** (Soundness of `CoVeNN`) *Let $\mathcal{N}$ be a neural network and $\phi_{in}, \phi_{out}$ be input/output properties such that `CoVeNN` verifies $\mathcal{N}$ satisfies $\phi_{in} \implies \phi_{out}$. Assume all local subproblems are formally verified by a sound underlying verifier $\mathcal{V}$ or formally refined as Thm. 2. Then $\mathcal{N}$ indeed satisfies $\phi_{in} \implies \phi_{out}$.*

*Proof.* Follow the arguments in §B.1, for any $x$ with $\phi_{in}(x)$, we have

$$\phi_{in}(x) \implies \gamma_1(\mathcal{N}_1(x)) \implies \gamma_2(\mathcal{N}_2(\mathcal{N}_1(x))) \implies \cdots \implies \phi_{out}(\mathcal{N}(x)).$$

Thus, $\mathcal{N}$ satisfies $\phi_{in} \implies \phi_{out}$. The correctness of `CoVeNN` is thus relative to the soundness of the underlying verifier: if the verifier establishes each local property, then the global specification holds. $\qquad\square$

The algorithm terminates because only finitely many refinement rounds are permitted for each $\gamma_i$. However, completeness is not guaranteed: the verifier may fail to establish some obligations or the refinement limit may be reached, in which case `CoVeNN` may output `unknown`.

# C   Constructing Assumptions

We seek to automate the generation of assumptions for the second obligation in Eq. 1, $I$, by adapting the verification of the first rule to approximate them. Alg. 2 outlines a systematic method to estimate and construct these assumptions. We use $\gamma$ to denote the values of intermediate steps in the process of computing $I - \gamma_{in}$ is the assumption for $\mathcal{N}$, and $\gamma_{assume}$ is the computed assumption for the subsequent network. The underlying verifier is able to compute a sound overapproximation of a given subnetwork's output for $\gamma_{in}$, which we denote with $\gamma_{over}$. These bounds, $(L_{over}, U_{over})$, are often too imprecise to allow verification of the overall problem.

We construct tighter assumptions, $\gamma_{assume}$, by interpolating between the overapproximation and a space defined by sampling the behavior of $\mathcal{N}$. Fig. 2b illustrates three distinct types of regions. The solid red line represents the *actual* operational region of a hidden ReLU within a DNN. Due to the inherent non-linearity of DNNs, calculating this exact region for hidden neurons is computationally infeasible. Instead, verification processes commonly use sound overapproximations to capture the behavior of each neuron, such as the triangular area bounded by $(L_{over}, U_{over})$. However, this introduces imprecision which can accumulate during verification of the layers of a DNN.

We compute a sampling region by minimizing (maximizing) the output of $\mathcal{N}$ for a set of sample inputs from $\gamma_{in}$ (line 2-line 3). This process uses backpropagation which requires that the minimization and maximization be performed separately [40]. These samples offer a tighter, more realistic representation of the network output bounds by considering adversarial conditions. The sampled region, $(L_{sample}, U_{sample})$, is likely to disprove subsequent verification sub-problems, since there is a high probability of counterexamples existing close to these bounds $L_{sample}$ and $U_{sample}$. To mitigate this risk, we compute an interpolated region, $(L_{int}, U_{int})$, between the sampled region and the overapproximation (line 4-line 5); the degree of interpolation can be controlled by $F$.

Rather than attempt to prove the full hypercube of an assumption, modern DNN verifiers [19, 8, 9, 10] have been engineered to prove subsets of clauses of the DNF encoding of the negation of the assumption – this is significantly faster in practice. We select the $P$ neurons with the largest interval size as determined by sampling to tighten (line 7). Selected constraints are then tightened by using the upper/lower bounds from $\gamma_{int}$ (line 8-line 9). Selective tightening allows us to mitigate the cost of downstream verification, since after the initial verification pass, only assumptions for tightened neurons need to be re-verified.

# D   Related Work

SoTA tools in DNN verification, such as those evaluated in VNN-COMP, integrating multiple techniques to improve scalability and efficiency. Leading verifiers, including `NeuralSAT` [24, 10] and $\alpha\beta$-`CROWN` [19, 8], split problems into smaller subproblems and refine bounds on subproblems. `NeuralSAT` and $\alpha\beta$-`CROWN` leverage GPU-accelerated linear bound propagation alongside advanced BaB techniques, such as cutting planes and neuron stabilizing, to handle harder networks. `Marabou` [11] encodes verification as constraint problems and employs parallelized split-and-conquer strategies to improve scalability. `nnenum` [12] achieves impressive performance on low dimensional networks using star sets and zonotope abstractions. `CoVeNN` can be viewed as a meta-verifier that decompose large verification problems into smaller subproblems, and leverage these existing SoTA verifiers to solve them.

There are two notable prior works addressing on compositional verification of DNNs. Ivanov et al. [23] introduces a compositional framework to break a high-level task into subtasks or subcomponents, such as breaking down car navigation task into track segment, each representing a distinct system mode (e.g., going straight or turning). Unlike `CoVeNN`, which focuses on decomposing the DNN itself, this work decomposes the task that may rely on DNNs.

`Prophecy` [22] uses an expensive decision tree algorithm to infer intermediate specifications. Their approach (1) is exponentially with the number of neurons, (2) records only activation patterns for ReLUs at one single layer (layer pattern) which is selected by hand, (3) does not support iterative refinements, and (4) and depends on training data. `CoVeNN` addresses these by inferring specifications by recording computed bounds for neurons which (1) scales along with DNN verifier algorithms, (2) records richer intermediate specifications that record ranges of activation values–which is essential for handling non-ReLU activations and automatically infers decompositions, (3) allows intermediate

specifications to be refined based on the property being checked, and (4) does not require training data. Lastly, `Prophecy` explicitly does not support Resnet and reports results on a few samples of small networks (e.g., ACAS Xu) which are not representative of modern DNNs.

# E   More Details on Experiments

## E.1   Experimental Setup, Solver Selection, and Benchmarks

**Setup**   Our experiments are conducted on a desktop with an AMD Ryzen Threadripper PRO 5975WX 32-Core, 128 GB RAM, and an NVIDIA GeForce RTX 4090 GPU with 24 GB VRAM. Our implementation is based on the open-source `NeuralSAT` verifier[3] with decomposition related code added. For RQ1 and RQ2, `NeuralSAT` was used as the backend verifier, while both $\alpha\beta$-`CROWN` and `NeuralSAT` were used in RQ3. Timeout for a single instance is set as 3600 seconds. The maximum number of rounds $r$ is set to 4. Note that these parameters can be changed easily by `CoVeNN`'s user.

**Underlying Solver Selection**   We tried several CPU-based verifiers, such as `nnenum` [12] and `Marabou` [11], but none of them could solve any instance of the benchmarks and their results would be all "-" if added to Tab. 2. In addition, we also tried running $\alpha\beta$-`CROWN` and `NeuralSAT` on CPU with instances that they got OOM errors and they all ended up being killed by the operating system or took more than 30 minutes even for a single abstraction pass. This is not surprising as our work is explicitly designed to solve problems that are beyond what can be solved by current NNV tools.

**Benchmark Selection**   The VNN-COMP benchmark suite is a useful starting point, but prior work [13, 41] notes that its benchmarks are often too easy for best verifiers. Our goal is to find benchmarks that go beyond what can be solved. We approached this from two directions: (1) scaling an existing VNN-COMP benchmark and (2) introducing a new challenging benchmark family.

Most VNN-COMP benchmarks lack a clear path for scaling, but the ResNet benchmark is an exception: increasing the number of residual blocks aligns with real-world models like DroNet [42] and ResNet-152 [20]. We used this scaling strategy, adapting the training setup from Pytorch Image Models (timm)[4].

To complement ResNet, we added a new benchmark family based on variational autoencoders (VAEs), which are naturally decomposable at their latent bottleneck layer. We adapted the encoder/decoder from Stable Diffusion[5], reduced its complexity for tool compatibility, and trained it on CIFAR10 using the simplified version of the original training code.

**Memory Usage**   We use information reported by CUDA to compute memory consumption, which represents the total memory demand/available memory. When the verifier attempts to allocate more memory than the system could provide, this exceeds 100% and causes OOM. In particular, we extract 3 memory metrics: current memory usage ($mem_{cur}$), free memory ($mem_{free}$), and memory requested for the operation that caused the OOM ($mem_{req}$). The memory usage calculation is:

$$mem_{usage} = \frac{(mem_{cur} + mem_{req} - mem_{free})}{mem_{total}} \qquad (10)$$

## E.2   Model architectures

We summarize the model structures in our experiments in Tab. 3. Let $Conv(a, b, c)$ be a conventional convolutional layer with $a$ input channel, $b$ output channels and a kernel size of $c \times c$. Let $ConvTran(a, b, c)$ be a transposed convolutional layer with $a$ input channel, $b$ output channels and a kernel size of $c \times c$. The stride and padding sizes are intentionally omitted for simplicity. Let $Lin(a, b, c)$ be a fully-connected layer with $a$ input features and $b$ output features. Let $ResBlock(a, b)$ stands for a residual block that has $a$ input channels and $b$ output channels. A $ResBlock$ is comprised of 2 paths, where the main path contains 2 $Conv$ and the residual path contains 1 $Conv$. All networks use ReLU activation only.

---

[3]https://github.com/dynaroars/neuralsat
[4]https://github.com/huggingface/pytorch-image-models
[5]https://github.com/explainingai-code/StableDiffusion-PyTorch

Tab. 3: Model architectures used in our experiments.

| Network | Architecture | Params |
|---------|--------------|--------|
| VAE_BASE | Encoder: $Conv(3, 8, 3), ResBlock(8, 8), Conv(8, 8, 4), ResBlock(8, 8), Conv(8, 1, 3), Conv(1, 1, 1)$
Decoder: $Conv(1, 1, 1), Conv(1, 8, 3), ResBlock(8, 8), ConvTran(8, 8, 4), ResBlock(8, 8), Conv(8, 3, 3)$ | 10K |
| VAE_WIDE | Encoder: $Conv(3, 16, 3), ResBlock(16, 16), Conv(16, 16, 4), ResBlock(16, 16), Conv(16, 1, 3), Conv(1, 1, 1)$
Decoder: $Conv(1, 1, 1), Conv(1, 16, 3), ResBlock(16, 16), ConvTran(16, 16, 4), ResBlock(16, 16), Conv(16, 3, 3)$ | 39K |
| VAE_DEEP | Encoder: $Conv(3, 8, 3), ResBlock(8, 8), 2 \times \left[ Conv(8, 8, 4), ResBlock(8, 8) \right], Conv(8, 1, 3), Conv(1, 1, 1)$
Decoder: $Conv(1, 1, 1), Conv(1, 8, 3), 2 \times \left[ ResBlock(8, 8), ConvTran(8, 8, 4) \right], ResBlock(8, 8), Conv(8, 3, 3)$ | 15K |
| RESNET6 | $Conv(3, 16, 3), ResBlock(16, 32), 05 \times ResBlock(32, 32), Lin(32, 10)$ | 113K |
| RESNET12 | $Conv(3, 16, 3), ResBlock(16, 32), 11 \times ResBlock(32, 32), Lin(32, 10)$ | 230K |
| RESNET18 | $Conv(3, 16, 3), ResBlock(16, 32), 17 \times ResBlock(32, 32), Lin(32, 10)$ | 348K |
| RESNET36 | $Conv(3, 16, 3), ResBlock(16, 32), 35 \times ResBlock(32, 32), Lin(32, 10)$ | 706K |

Tab. 4: Number of completed jobs for each verifier (**verified**, **unknown**); "-" means that the verifier aborted with an OOM error on all 10 properties.

| #Blocks | $\alpha\beta$-CROWN | NeuralSAT | CoVeNN$_{NS}$ |
|---------|---------------------|-----------|----------------|
| 3 | (6, 4) | (6, 4) | (6, 4) |
| 6 | (5, 5) | (5, 5) | (5, 5) |
| 9 | (8, 2) | (9, 1) | (9, 1) |
| 12 | - | - | (7, 3) |
| 18 | - | - | (8, 2) |
| 36 | - | - | (6, 4) |

## E.3  Tools Scalability

Tools' scalability of the two top performing verifiers in VNN-COMP'24, $\alpha\beta$-CROWN and NeuralSAT, and CoVeNN on ResNet-based instances is shown in Tab. 4. Number of completed jobs for each verifier is (**verified**, **unknown**) where "-" means that the verifier aborted with an OOM error on all 10 properties.

As detailed in Tab. 4, CoVeNN matches the ability of underlying verifiers on problems for which they complete and the refinement strategies allow it to prove properties when scaling to much larger networks.

