# OpenReview forum: "Compositional Neural Network Verification via Assume-Guarantee Reasoning"
_NeurIPS.cc/2025/Conference — NeurIPS 2025 spotlight_

### Official Review · Reviewer_GNvQ · 2025-06-02

**Clarity:** 3
**Significance:** 2
**Originality:** 3
**Rating:** 3
**Confidence:** 4

**Summary:**

The paper proposes a compositional neural network verification approach for reducing the memory load of existing verification algorithms. Therefore, the computation graph of a neural network is split into parts that are incrementally verified using intermediate assumptions derived from heuristics; the verification of each part ensures that its output entails the assumption of the successor part. The intermediate assumptions can be iteratively refined. The evaluation demonstrates that the proposed approach can improve state-of-the-art verification algorithms when the available memory is limited.

**Questions:**

- Does the approach have other benefits than reduced memory load for large neural networks?
- How are the computation graphs of the neural networks decomposed? Could you please elaborate with an example?
- How large is the conservatism introduced by the intermediate assumptions? How are the intermediate assumptions represented, e.g., using intervals?
- Why are the evaluations only done on non-standard benchmarks? There are many instances in the VNN-COMP that are not solved, e.g., a-b-CROWN only solves 74.5% of the CIFAR100 benchmark?
- How does the approach affect computation time?

**Ethical Concerns:**

["NO or VERY MINOR ethics concerns only"]

**Final Justification:**

While the topic of handling relatively large networks under memory constraints is interesting, the applied techniques do not seem novel to me. Paired with missing description on how the models as decomposed (via which heuristics?), leads me leaning towards reject although I can also see it being accepted.

**Limitations:**

yes

**Paper Formatting Concerns:**

-

**Quality:**

2

**Strengths And Weaknesses:**

Strengths:
- The paper is well-written and the approach is well explained.
- Exploring how large networks can be verified is an interesting research direction.
- The claimed improvement in memory efficiency is demonstrated in the evaluation.

Weaknesses:
- The related work section is very brief and should be extended.
- The paper does not go into the details on the heuristics for splitting the computation graph of the neural networks.
- The proposed approach only benefits the verification of large neural networks with limited memory.
- The evaluation is done on non-standard benchmarks.

---

> ### Author Rebuttal · Authors · 2025-07-30
>
> Thank you for your critical evaluation. Please find our responses below.
>
> 1. Other benefits than reduced memory
>
> Beyond memory savings, CoVeNN doesn't give many other benefits. However, as mentioned by reviewer qN2F, the intermediate bounds computed during decomposition can be reused for solving multiple objectives of the same problem or save computation across different verification tasks.
>
> 2. Networks decomposition and example
>
> CoVeNN decomposes networks by identifying strategic "cuts" in the computation graph that divide the network into sequential subnetworks (Sect.3.1).
>
> Consider a network in our evaluation ResNet-36 with 110 layers. CoVeNN would decompose it as follows:
> - Subnetwork N_1: Input layer + first CNN + first 6 ResNet blocks
>     - Input: Original image constraints (robustness)
>     - Output assumption γ_1: Interval bounds on 32x32x32 feature maps
> - Subnetwork N_2: Next 6 ResNet blocks
>     - Input assumption: γ_1 (refined)
>     - Output assumption γ_2: Interval bounds on 32x32x32 feature maps
> - Subnetwork N_3: Final 24 ResNet blocks + classification head
>     - Input assumption: γ_2 (refined)
>     - Output: Final classification property (e.g., argmax(output) = true_label)
>
> 3. Intermediate assumptions conservatism and representation
>
> The conservatism (abstraction) varies based on network architecture, property being checked, and the refinement process.  Currently we use polytopes to compute bounds, and CoVeNN goes through refinement processes to tighten bounds.
>
> Yes, CoVeNN represents intermediate assumptions at the cut point using hyperrectangular interval bounds. CoVeNN can use other representations but we use intervals in this work for compatibility with underlying verifiers, which typically work with interval preconditions.
>
> 4. Applied to VNN-COMP instances that cannot be solved
>
> The VNN-COMP instances that tools such as a-b-CROWN cannot solve are not due to scalability (e.g., memory limited) but rather due to imprecise abstraction employed by the verifier. CoVeNN and decomposition in general won't be able to help as such problems are better solved with better abstraction and smarter branching/decision heuristics.
>
>
> 5. How does the approach affect computation time?
>
> CoVeNN first runs a verifier to solve the problem and only invokes decomposition if the verifier fails. Thus if no decomposition is required then the overhead is 0 and decomposition only incurs overhead when the problem cannot be solved monolithically.

---

### Official Review · Reviewer_dCtJ · 2025-06-30

**Clarity:** 2
**Significance:** 3
**Originality:** 4
**Rating:** 5
**Confidence:** 3

**Summary:**

This paper proposes a novel approach to mitigate the high memory consumption involved in verifying large neural networks. The method decomposes a single verification problem into a sequence of subproblems, each of which can be handled by an existing verifier with lower memory requirements. Empirical results show that the proposed technique enables the verification of larger network architectures while reducing overall memory usage.

**Questions:**

1. Why is the decomposition method not applied to disproving properties? I would be happy to raise my score if the authors can provide a clear justification. It would also be helpful to include this clarification in the main paper.
2. Why does the memory usage exceed 100% in Figure 3b?

**Ethical Concerns:**

["NO or VERY MINOR ethics concerns only"]

**Final Justification:**

The authors provide a justification and an idea for addressing the incompleteness of their current method. I am satisfied with their response and have increased my score accordingly.

**Limitations:**

The limitations section does not mention the method’s inability to disprove properties, as noted in the Quality section.

**Paper Formatting Concerns:**

N.A.

**Quality:**

3

**Strengths And Weaknesses:**

### Quality
The approach appears sound to me. The authors present correctness theorems along with proof sketches in the main paper, although I did not review the full proof provided in the appendix. The empirical results, particularly those in Table 2, clearly demonstrate the proposed method’s superior ability to handle large-scale neuron networks compared to existing state-of-the-art techniques.

However, my primary concern is the limitation of this approach in disproving properties. The proposed method can only verify that a property holds, but it cannot refute a property by providing a counterexample. As a result, the empirical evaluation only focuses on networks with valid properties. This is a notable weakness compared to most existing verifiers, such as $\alpha\beta$-CROWN and NeuralSAT, which can produce counterexamples when a property does not hold.

### Clarity
The paper is generally easy to follow. However, too many algorithmic details are deferred to the appendix. For instance, the roles of the input parameters P and F in Algorithm 1 are not clearly specified, and the generateAssumption function (line 10) lacks sufficient explanation.
To improve clarity, the authors could consider including more of these important implementation details in the main text. Given the page limit, this could be achieved by reducing the space of the formal correctness section, which, in my view, does not contribute significant theoretical results.

### Significance
The improvement in verifying large neural networks with valid properties is compelling; however, the inability to disprove properties makes the proposed method incomplete.

### Originality
This divide-and-conquer-based approach to neural network verification appears novel and promising. With further development, it has the potential to effectively address the issue of high memory consumption in neural network verification.

---

> ### Author Rebuttal · Authors · 2025-07-30
>
> Thank you for your time and thoughtful comments. Please find our responses below.
>
> 1. Disproving properties
>
> This is a thoughtful question. The short answer is that decomposition might be able to disprove properties, however we haven’t explored this direction because verifiers rarely have issues with disproving properties (their employed adversarial attacks are highly effective at finding counterexamples), and existing benchmarks from VNN-COMP contain virtually no SAT (disproving) cases that cause scalability issues, giving us no opportunity to evaluate compositional disproving.
>
> That said, we anticipate that a challenge in using compositional disproving is due to the nature of over-approximation that favors verification over falsification.  Counterexamples computed from overapproximation may not be valid and require additional checking and refinement. That said, by answering this question, we came up with an idea for compositional disproving by adapting preimage methods (e.g., PREMAP: A Unifying PREiMage Approximation Framework for Neural Networks - Xiyue Zhang et. al.). These backward analysis techniques find under-approximations of inputs reachable to predefined output spaces. Starting from a violated output space, we could compute intermediate spaces reachable to that output space through backward analysis until reaching the input precondition. If the found preimage space is a subset of the precondition, we can confirm the violation.
>
> 2. Memory usage exceeds 100%
>
> We use information reported by CUDA to compute this, which represents the total memory demand / available memory.  When the verifier attempts to allocate more memory than the system could provide, this exceeds 100% and causes OOM.
>
> In particular, we extract 3 memory metrics: current memory usage (mem_cur), free memory (mem_free), and memory requested for the operation that caused the OOM (mem_req). The memory usage calculation is:
>
> mem_usage = (mem_cur + mem_req - mem_free) / mem_total

---

> > ### Comment · Reviewer_dCtJ · 2025-08-06
> >
> > Thank you for your response.
> >
> > From a practical standpoint, it is essential to have a complete method, as we cannot determine the satisfiability of an instance in advance. An incomplete tool may hang on a potentially satisfiable instance, which limits its usability. Therefore, I strongly encourage you to include your justification and any new ideas on disproving properties in the main paper to further strengthen your work.
> >
> > I have increased my score accordingly.

---

> ### Author Response · Authors · 2025-08-06
>
> Thanks! We will add a discussion on disproving properties and include ideas on doing so (e.g., using pre-imaging and backward analysis as mentioned in our original rebuttal).

---

### Official Review · Reviewer_2aSz · 2025-07-01

**Clarity:** 3
**Significance:** 2
**Originality:** 3
**Rating:** 5
**Confidence:** 4

**Summary:**

This paper introduces CoVeNN, a framework for verifying DNNs using assume-guarantee compositional reasoning. CoVeNN decomposes large networks into smaller subnetworks, verifies them independently with an off-the-shelf verifier, and composes the results to establish the property for the whole network. The framework includes an iterative refinement strategy that tightens assumptions to reduce overapproximation. Experiments on VAE and ResNet benchmarks show that CoVeNN can verify significantly larger networks than sotaf verifiers, solving 6–7 times more properties without running into GPU memory issues.

**Questions:**

- The decomposition strategy assumes linear, sequential splits. How would CoVeNN handle non-linear topologies like RNN?

- The refinement loop is heuristic and may return unknown if bounds don’t converge. Is there any theoretical or empirical guarantee that refinement will succeed within the allowed rounds? If not, how can users avoid wasted compute?

- The paper relies on heuristics for cuts, neuron selection, and scaling factors. How robust is CoVeNN to poor hyperparameter choices, and have you explored automated tuning?

- The method depends on verifiers that assume piecewise-linear activations. How does CoVeNN generalise to networks with smooth or non-linear activations (e.g., GELU, tanh, sigmoid)? Does the soundness or refinement extend naturally to these cases?

**Ethical Concerns:**

["NO or VERY MINOR ethics concerns only"]

**Final Justification:**

The authors responses and clarifications, both to myself and to the other reviewers, have satisfactorily addressed the concerns and questions I had regarding the paper.  Therefore, I lean towards acceptance.

**Limitations:**

Yes

**Quality:**

3

**Strengths And Weaknesses:**

Strengths:
- Clear motivation: It’s a well-motivated practical problem, especially as DNNs continue to grow in depth and complexity.
- Novelty: It introduces compositional techniques, specifically, assume-guarantee reasoning from formal verification, and adapts them for DNN verification. While the way of using assume-guarantee reasoning is standard as in formal verification, its application here in a concrete, verifier-independent manner appears for DNN, to the best of my knowledge, to be a first.
- Strong empirical evidence: The experiments show that CoVeNN can verify significantly larger networks than the standalone underlying verifiers can.


Weaknesses:
- Limited adaptability to more complex architectures: The proposed decomposition assumes a mostly sequential, acyclic structure and may not naturally extend to models like transformers, RNNs, or GNNs, where the computation graph can be inherently cyclic or non-linear. For example, how would the decomposition handle genuine cyclic dependencies in RNNs, where the hidden state feeds back recursively?
- No guarantees for refinement convergence: The iterative refinement loop is heuristic and may fail to converge. The paper does not provide strong theoretical or empirical guarantees explaining when or why convergence will succeed within the allowed rounds.

---

> ### Author Rebuttal · Authors · 2025-07-30
>
> Thank you for your time and thoughtful comments. Please find our responses below.
>
> 1. Handling non-linear topologies like RNN
>
> The sequential composition we describe in the paper is one common case, but the framework supports more complex acyclic topologies by finding appropriate "cuts" in that graph. For RNNs, we believe CoVENN can handle them through unrolling, a common strategy in software verification. By fixing the number of recurrent steps and unrolling the RNN into an equivalent feed-forward network, we can apply CoVeNN's decomposition strategy. Existing work in RNN verification (e.g., Verifying Recurrent Neural Networks using Invariant Inference - Yuval Jacoby et. al.) applies unrolling before verification. Interestingly, unrolled RNNs create very natural decomposition points after each unrolled timestep, making them well-suited for our compositional approach.
>
> For more general cyclic computation graphs, handling cycles would require incorporating more sophisticated circular assume-guarantee proof rules (e.g., Automated circular assume-guarantee reasoning - KA Elkader et. al.).
>
> 2. Refinement loop may return unknown
>
> Indeed the refinement can return unknown, and we do not believe there is any guarantee  that refinement will succeed within the allowed rounds. Currently, wasted computation does occur in our approach: after reaching the number of allowed rounds, the process will stop and return unknown.
>
> However, there could be empirical early stopping strategies to avoid wasted computation. For example, in VAEs where the dimension of decomposed layers is generally small, CoVeNN may refine assumptions of the same dimensions multiple times across rounds. If the refinement yields no improvement on any dimension, we can stop immediately since the current settings (timeout for verifying each assumption, etc.) won't allow CoVeNN to improve further.  We will try this in future work.
>
> 3. Parameter choices and automated tuning
>
> We included the detailed ablation study of parameters on the appendix E.3.2, L289. The algorithm indeed is affected by neuron selection (P) and less sensitive to scaling factor (F).   We haven’t yet explored automated tuning.
>
> 4. Non-linear activations (e.g., GELU, tanh, sigmoid)?
>
> CoVeNN, and in general DNN verification tools, are not limited to piecewise-linear activations and can handle smooth or non-linear activations, e.g., tanh, sigmoid, etc. The key insight is CoVeNN represents intermediate assumptions using interval bounds at the cut point rather than discrete neuron status (on/off), making it activation-agnostic. CoVeNN captures over-approximated lower and upper bounds for each neuron at intermediate layers, and these interval bounds remain valid regardless of the activation function used. When refining these bounds, the refined intervals remain over-approximated, ensuring soundness.

---

> > ### Comment · Reviewer_2aSz · 2025-08-06
> >
> > I thank the authors for their detailed rebuttal. The responses have addressed the concerns and questions I had regarding the paper. I have no further concerns and, accordingly, will increase my final score.

---

> > > ### Author Response · Authors · 2025-08-07
> > >
> > > Thanks. We appreciate the review and feedback, and will discuss your comments in the paper revision

---

### Official Review · Reviewer_qN2F · 2025-07-01

**Clarity:** 4
**Significance:** 4
**Originality:** 4
**Rating:** 6
**Confidence:** 4

**Summary:**

The paper proposes CoVeNN, a methodology for the compositional verification of neural networks.
The approach first decomposes a neural network into multiple blocks and then verifies these blocks individually
in an assume-guarantee manner. If a property is not verifiable, the approach refines the intermediate assumptions
to achieve verification.
To this end, a number of heuristics are proposed.

**Questions:**

- **(O1)** Regarding (O1): The coarse over-approximation certainly provides guarantees for a much wider part of the input space than what is verified for a given robustness property.
		  Have you inspected the inner contracts? Maybe they could be reused? Especially for image NNs it is also conceivable,
		  that it might be possible to derive some interpretable information from them (something like if these nodes are in range [l,u] the NN is recognizing a wheel of a car)
- **(O2)** It is commonly believed that the internal representations of ResNets and VAEs correspond to higher-level features of the provided inputs (e.g. image components) which seems to be
  particularly well suited for compositional verification. Have you made any observations in this direction? Do you expect this approach to work equally well for other NN architectures?
- **(O3)** On a more practical level: Do you have any indication whether compositional verification works so well "only" because it yields significant memory savings
  (i.e. given larger RAM NeuralSat could solve the same problem) or does it also provide a significant benefit in terms of computation time (i.e. even with more memory NeuralSAT would be much slower)?

**Ethical Concerns:**

["NO or VERY MINOR ethics concerns only"]

**Final Justification:**

Overall, I find this paper to be novel, interesting, and a step forward in handling memory-intensive verification tasks.
I found the authors' answers to be satisfactory and hence strongly believe this paper should be accepted for the reasons outlined in my review above.
In particular, I believe there are good reasons why the paper does not evaluate on VNN COMP benchmarks as outlined in the rebuttal to GNvQ.
Moreover, verification of recursive architectures is a highly complex topic of its own, and I view this approach as significant independently of whether it can be generalized to such architectures (although the authors even claim it can be extended in this direction in the rebuttal to  2aSz).

**Limitations:**

yes

**Quality:**

4

**Strengths And Weaknesses:**

**Strengths:**
- **(S10)**	The novel, innovative approach convincingly demonstrates that compositional verification for neural networks is A) achievable
		and B) increases the scale of NNs that can be processed.
- **(S20)**	The approach is agnostic to the underlying verifier and can thus scale as NN verification technology develops further.
		This is particularly exciting, because it hints at the fact that speedups from further improvements in NN verification technology
		might multiply with the speedups achievable via decomposition (rather than these effects cancelling each other out).
- **(S30)**	The evaluation demonstrates that the approach can solve problems that are much larger than what the current State of the Art can achieve.
- **(S40)**	The paper does a good job explaining the approach and provides helpful illustrations and examples.


**Minor Opportunities for Improvement:**
- **(O1)**	I am under the impression that work on shared certificates in NN verification [A] is another related technique that might warrant discussion in Appendix D.
		While the approach has a slightly different objective (reusing contracts for verification of other properties), it also leverages assume-guarantee reasoning about intermediate values.
		Moreover, it might be interesting to combine these techniques in future work: Maybe derived guarantees about intermediate blocks by COVeNN could equally be reused for other (robustness) properties?
- **(O2)**	The approach's performance is still dependent on a number of hyperparameters which must be tuned correctly. However, this is also transparently discussed by the authors.
		While this is a limitation of the appraoch at the current stage, I believe the major contribution of the paper is to demonstrate how compositional verification can, in general, be achieved.
		This contribution is not lessened by the need for more engineering concerning these hyperparameters.

**Minor:**
- Line 82f: Typo in the formula: I belive it should read $\max\left( [...] \cdot v_{i-1,k} [...] \right)$ (replace j with k)
- Line 210: The text seems to indicate, that the interpolation is happening with $s=0.5$, but the numbers of $\gamma_1'$ do not seem to match a 0.5 interpolation between $\gamma_1$ and $\sigma_1$.
- Line 262: "none of them are sat" -- is it known that the unknown benchmarks are unsatisfiable or do we simply not know their status yet, because no tool solved them so far?

[A] Fischer, Marc, et al. "Shared certificates for neural network verification." International Conference on Computer Aided Verification. Cham: Springer International Publishing, 2022.

---

> ### Author Rebuttal · Authors · 2025-07-30
>
> Thank you for the encouraging review and thoughtful comments. Please find our responses below.
>
> 1. Reuse intermediate assumptions and interpretability
>
> Indeed the computed over-approximation can be reused, and we exploited a bit of that in our implementation (for the same precondition but different postconditions, e.g., a robustness property on an image with the same input constraints but different output specifications like disjunctions over multiple output classes, we avoid recomputing intermediate bounds). We haven’t explored more general cases as you suggested and will do so in future work,  e.g., after verifying a property with intermediate bounds [l, u] at a given layer, these verified contracts can be cached and reused for future verification tasks where the intermediate bounds are subsets of [l, u], providing significant computational savings.
>
> While we haven't considered interpretability to date, your question suggests an approach as follows. For image classification networks like ResNets, the intermediate bounds computed by CoVeNN at different layers could potentially correspond to semantic features (e.g., edges, textures, object parts). The compositional nature of our approach naturally aligns with the hierarchical feature learning in CNNs, where early layers capture low-level features and deeper layers capture high-level semantic concepts. By inspecting these intermediate contracts, perhaps using concept activations, we could potentially gain insights into which feature representations are being verified.
>
>
> 2. Internal representation of ResNets/VAEs well-suited for decomposition
>
> Yes, we observe that for VAEs, the bottleneck layers create natural bottlenecks where intermediate assumptions can be compactly represented, leading to more effective decomposition. For ResNets, however, the observation is not as clear due to their large internal dimensions (32x32x32) which requires CoVeNN to select only a subset of neurons for refinement. Our current selection strategy depends solely on bound tightness, but selecting neurons using heuristics that reflect neuron-feature associations might improve solving capability further.
>
> We expect CoVeNN to work well on other architectures, though the degree of benefit may vary. In particular, Diffusion Models (or unrolled RNNs) where a model is executed in K steps,  CoVeNN would be able to decompose the problem where existing verifiers might not be able to handle large K.
>
>
> 3. Improve solving time
>
> CoVeNN works well primarily because of memory savings and might not improve solving time. When the underlying verifier can successfully solve a problem, decomposition is less useful because it would introduce overhead in decomposition, refinement, and multiple rounds of verification

---

> > ### Comment · Reviewer_qN2F · 2025-08-03
> >
> > Thank you for the rebuttal which answered all of my questions.
> > I will keep my score.

---

> > > ### Author Response · Authors · 2025-08-07
> > >
> > > Thanks! Once again we appreciate the review and feedback, and will discuss them in the paper revision

---

### Note · Authors · 2025-08-13

We want to once again thank the reviewers for their time and constructive feedback---both critical and positive. Through the rebuttal and discussion, we were able to perform additional experiments to provide more evidence supporting the work or answering reviewers' questions. We believe we have addressed all main questions and confusions, and appreciate that the reviewers have taken our responses into their consideration and raising their scores accordingly.  While there might still be few reservations, e.g., CoVeNN was not designed to solve hard problems due to, e.g., abstraction imprecision (but rather using compositional approach to breakdown large problems that are beyond the reach of existing verifiers), we hope the reviewers find the changes and additional results satisfactory and, like us, are convinced that the work is a significant step toward improving DNN verification.

---

### Decision · Program_Chairs · 2025-09-17

**Decision:**

Accept (spotlight)

**Comment:**

(a) Summary of Scientific Claims and Findings

The paper presents CoVeNN, a framework that applies compositional "assume-guarantee" reasoning to make neural network verification (NNV) scalable. By decomposing large networks into smaller parts and verifying them independently, CoVeNN primarily addresses the memory limitations of existing verifiers. The authors claim this approach dramatically extends the scale of verifiable networks. Their findings show that CoVeNN solved nearly 7 times more problems on large VAE and ResNet models than state-of-the-art verifiers, which frequently failed due to out-of-memory errors.

(b) Strengths of the Paper

The paper's primary strengths were consistently praised by reviewers:

1. It introduces a novel and groundbreaking application of compositional reasoning to NNV, tackling the critical challenge of scalability.

2. It provides strong empirical evidence of scaling verification to networks far larger than what current tools can handle, a major practical advancement.

3. The framework is modular and can be used with any underlying verifier, allowing it to benefit from future improvements in NNV technology.


4. The complex approach is well-explained and illustrated, making it easy to follow.

(c) Weaknesses of the Paper

Initial reviews identified several limitations:

1. The method can verify properties but cannot find counterexamples to disprove them, a key feature of existing verifiers.

2. The decomposition strategy is primarily designed for acyclic networks, raising questions about its applicability to models with recurrent or cyclic structures.

3. The framework's performance depends on heuristics and hyperparameter tuning, and its iterative refinement process lacks convergence guarantees.

(d) Reasons for Acceptance

Despite its limitations, the paper earned a consensus for acceptance. Reviewers were highly impressed by the authors' rebuttal, which satisfactorily addressed their main concerns. The paper's core contribution—demonstrating a viable path to scale NNV to previously intractable models—was considered groundbreaking and significant enough to outweigh the current weaknesses. The authors also proposed credible plans to address the limitations in future work, reinforcing the reviewers' positive final assessments.


(e) Summary of Discussion and Changes

The rebuttal period was highly effective and pivotal to the paper's acceptance. Key issues were successfully resolved:

1. The authors acknowledged this limitation and proposed a concrete future research direction using backward analysis to find counterexamples, satisfying the reviewer's concern.

2. They clarified that the method could handle RNNs by "unrolling" them into feed-forward networks, a standard practice that makes them suitable for the compositional approach.

3. The use of custom-scaled benchmarks was justified by the need to create problems that are challenging enough to show the limitations of existing verifiers, which was the paper's goal.

The authors' thorough and convincing responses led to multiple reviewers increasing their scores and solidifying their recommendations for acceptance.